# Online biology degree program broadens access for women, first-generation to college, and low-income students, but grade disparities remain

Chris Mead[1,☯], K. Supriya[2,☯], Yi Zheng[3], Ariel D. Anbar[4], James P. Collins[5], Paul LePore[6], Sara E. Brownell[2]*

1 Center for Education Through Exploration, School of Earth and Space Exploration, Arizona State University, Tempe, Arizona, United States of America, 2 Biology Education Research Lab, Research for Inclusive STEM Education Center, School of Life Sciences, Arizona State University, Tempe, Arizona, United States of America, 3 School of Mathematical and Statistical Sciences, Mary Lou Fulton Teachers College, Arizona State University, Tempe, Arizona, United States of America, 4 Center for Education Through Exploration, School of Earth and Space Exploration, School of Molecular Sciences, Arizona State University, Tempe, Arizona, United States of America, 5 School of Life Sciences, Arizona State University, Tempe, Arizona, United States of America, 6 College of Liberal Arts and Sciences, Arizona State University, Tempe, Arizona, United States of America

☯ These authors contributed equally to this work.
* Sara.Brownell@asu.edu

## Abstract

Online education has grown rapidly in recent years with many universities now offering fully online degree programs even in STEM disciplines. These programs have the potential to broaden access to STEM degrees for people with social identities currently underrepresented in STEM. Here, we ask to what extent is that potential realized in terms of student enrollment and grades for a fully online degree program. Our analysis of data from more than 10,000 course-enrollments compares student demographics and course grades in a fully online biology degree program to demographics and grades in an equivalent in-person biology degree program at the same university. We find that women, first-generation to college students and students eligible for federal Pell grants constitute a larger proportion of students in the online program compared to the in-person mode. However, the online mode of instruction is associated with lower course grades relative to the in-person mode. Moreover, African American/Black, Hispanic/Latinx, Native American, and Pacific Islander students as well as federal Pell grant eligible students earned lower grades than white students and non-Pell grant eligible students, respectively, but the grade disparities were similar among both in-person and online student groups. Finally, we find that grade disparities between men and women are larger online compared to in-person, but that for first-generation to college women, the online mode of instruction is associated with little to no grade gap compared to continuing generation women. Our findings indicate that although this online degree program broadens access for some student populations, inequities in the experience remain and need to be addressed in order for online education to achieve its inclusive mission.

**Data Availability Statement:** All relevant data are within the manuscript and its Supporting Information files.

**Funding:** This work was supported by grant #GT11046 from the Howard Hughes Medical Institute (www.hhmi.org), awarded to JPC, SEB, PL, and ADA and grant #1711272 from the National Science Foundation (www.nsf.gov), awarded to SEB. The funders had no role in study design, data collection and analysis, decision to publish, or preparation of the manuscript.

**Competing interests:** The authors have declared that no competing interests exist.

# Introduction

Online college education has been growing rapidly, not only in terms of the number of students enrolled in online courses and the number of courses offered, but also in the number of fully online degree programs. About 33% of college students take online courses and 15% of college students are enrolled in fully online degree programs, including 7.3% of students in 4-year undergraduate degree programs [1, 2].

Because fully online education removes many of the constraints imposed by a brick and mortar location, it has enormous potential to increase access to students who may otherwise not be able to pursue a college degree [3]. First, online education could increase access for higher education for students who live in education deserts. Geographical inequalities in access to resources such as high quality K-12 public education and nutritious food along the lines of race and class are well documented in the US [4–6]. Similar spatial inequalities in access to higher education have been termed "education deserts" and are defined as places with no colleges or public universities within 25 miles or with access to only one community college as the only public broad access institution within 25 miles [7, 8]. According to one estimate, about 17.6% of American adults live in an education desert [8]. Akin to other forms of structural inequalities in our society such as food deserts, these education deserts are more common in low-income communities [7, 9]. While online education cannot entirely solve these inequities in access to educational opportunities because of the availability of the technology required for online degrees in some areas, many of the students who live in education deserts do have sufficient internet service to access online education [8]. A second way that online education can increase access is by accommodating a broader range of schedules. Asynchronous online education could provide college access for students with severe time constraints that necessitate a flexible course schedule, such as students who serve as primary caregivers (which are often women with childcare responsibilities) and those students who are seeking a degree while working a full-time job. Several studies comparing the demographics of students in online degree programs to in-person degree programs show that online programs indeed increase access for such student populations. Students in fully online degree programs are more likely to be women, older than students attending a 4-year residential college, and eligible to receive federal financial aid (e.g., Pell grants), although they are as or more likely to be white [10–15].

In addition to broadening access to higher education for students, online education could help meet the demand for a larger and more diverse STEM workforce. The President's Council of Advisors on Science and Technology (PCAST) has called for increasing the number of STEM majors by one million over the next decade [16]. Greater access to STEM degrees through online programs could contribute significantly to meeting this goal, especially because many of our brick and mortar campuses are limited in their capacity to offer physical classes and thus limited in their student enrollment [17]. Although most online degree programs currently offered are in non-STEM disciplines, fully online degree programs constitute 14% of all health professions and related associate's or bachelor's degree programs in the sciences (2041 out of 14578 programs) [18]. For STEM disciplines, online degree programs are more common in fields such as computer and information sciences (21%), and the numbers are considerably smaller for biological and biomedical sciences, where fully online degree programs represent only 1% of associate's or bachelor's degree programs (33 out of 3300 programs) [18]. We know of only three bachelor's degree programs in biology from research intensive institutions that are fully online: University of Florida, Florida International University, and our program at Arizona State University. Further, only Arizona State University offers a Bachelor of Science degree.

It has been argued that online education's route to increased access to higher education could also amplify the diversity of the STEM workforce [19, 20]. A diverse STEM workforce composed of individuals with different backgrounds, perspectives, and identities can help make science and technology research and their applications more just in terms of what questions are examined, who is most likely to benefit, and who could potentially be harmed. Historically, the benefits of science and technology have been distributed disproportionately to more privileged groups in our society (e.g., disparities in federal funding between diseases such as Cystic Fibrosis that more often affects white people compared to sickle cell disease that more often affects people with sub-Saharan African ancestry [21]), while harms have predominantly fallen on people of color (e.g., unethical medical experimentation on Black Americans [22]). Greater participation in the STEM workforce across gender, race/ethnicity, and social class would contribute positively to social justice in STEM, increase objectivity in science by including diverse perspectives, and overall lead to higher quality science and technology [23, 24].

Women, Black students, Latinx students, Native American students, and students with low socioeconomic resources continue to be underrepresented in most STEM undergraduate degree programs [25, 26]. Online education seems to increase the diversity of students in STEM undergraduate degree programs to some extent. Studies show that women and students who grew up in low-income households constitute a larger proportion of students in STEM online courses as well as in fully online STEM degree programs. In a comparison of 174 online and 440 in-person biology students at a large public research-intensive (R1) institution in the southwestern USA, Cooper et al. [15] found that a greater proportion of online students were first-generation college students, grew up in a low-income household, worked more than 21 hours a week, and identified as primary caregivers. Other studies using the large NCES National Postsecondary Student Aid Study 2008 dataset document that the likelihood of taking an online STEM course was higher for women and students with non-traditional characteristics (e.g. single parents, full-time workers), but lower for Black and Hispanic/Latinx students even after taking other demographic variables into account [27, 28]. Thus, the current data suggest that online degree programs offer a way for a greater number of women and low-income students to pursue science, but there is no existing evidence that it can help racial or ethnic representation. This is potentially worrisome because there are large racial/ethnic disparities in science, particularly biology, and fully online degree programs may not address these racial/ethnic inequities.

Further, online degree programs may not increase the number of STEM graduates because student enrollment in online coursework is not equivalent to completion. Studies report 10–15% higher withdrawal rates from online courses compared to in-person courses in community colleges, even after controlling for students' prior academic achievement and other demographic variables [29, 30]. It is important to note that all of the studies referred to here have been examining courses for credit within a degree program; this does not include massive online open courses (MOOCs), which are well-known to have extremely low completion rates [31]. Another factor to consider is student performance in online courses. While several studies report comparable learning outcomes for students in online courses compared to face-to-face courses [14, 32], other studies show worse student performance in online courses [11, 12, 33, 34]. Xu & Xu [18] suggest that these differences in results might be explained by the selectivity of institutions studied. Studies done on community college systems and non-selective institutions consistently demonstrate negative student learning outcomes online, whereas studies at selective institutions tend to have mixed results [18]. Differences in the population of students who enroll in online courses at selective compared to non-selective institutions in terms of prior academic preparation and/or availability of time or resources might contribute

to this observed pattern. Some studies show that Black students [35, 36], Latinx students [36, 37], and low-income students [38] do worse in online courses compared to face-to-face courses and students with these social identities constitute a larger proportion of the student body in non-selective institutions [39].

Whether student performance in online courses is comparable to in-person courses may depend on the academic field examined. Some research has found that grade differences between online and in-person courses are smaller in the natural sciences than other fields [11, 35, 36]. In contrast, a comparison of retention rates for online and in-person courses taught by the same instructor in the same semester showed lower retention rates in STEM courses, particularly elective courses, compared to non-STEM courses for online students [40]. However, numerous studies have reported gender, racial, and socioeconomic disparities in grades received by students in in-person STEM courses and shown that men, white, and middle-or upper class students tend to receive higher grades compared to women, students of color, and low income students, respectively [41–47]. Because there is a greater degree of anonymity in participation in online courses, it could be hypothesized that students could experience less discrimination and stereotype threat in online courses, so students with social identities underrepresented in STEM could do better in online courses. However, the evidence so far does not support this idea. Wladis et al. [30] showed that women are more likely to get a grade equal to or higher than a C compared to men in face-to-face STEM courses, but that both men and women have similar probability of getting a grade equal to or higher than C in online courses, even after controlling for other confounding variables such as race/ethnicity, age, and prior GPA. However, the lack of gender disparity in grades received in online courses observed in this study was a result of women doing significantly worse in the online courses compared to in-person courses [30]. While Latinx and Black students generally received lower grades in STEM courses compared to white and Asian students, this grade gap has been shown to be similar in online and face-to-face courses, so previous studies suggest that the online format does not seem to lessen these grade disparities [30].

While there have been many studies of student performance in online courses compared to in-person courses, very few studies have examined student performance in fully online vs in-person degree programs. Fully online asynchronous degree programs offer much greater geographic and temporal flexibility than programs where students can take some online courses within the purview of an in-person degree program. Student experiences in such fully online programs likely differ significantly from student experiences in online courses nested within in-person degree programs, potentially because students enrolled in online programs tend to be older and have more work experience. Some of us published a study on students that completed a fully online astrobiology course that showed that students in the fully online degree program began the course with a higher sense of personal value of science compared to students in the in-person degree program [48]. We also conducted a study demonstrating that students in a fully online biology degree program who wanted to become doctors were more confident about their career choice, but knew less about the criteria that medical schools consider in their admissions process, compared to students in an equivalent in-person biology program [15]. Thus, beyond just learning outcomes and course grades, there could be downstream effects for students in online degree programs that influence their career trajectory. However, in order to begin to understand these effects, we argue that we first need to establish how student representation and student grades differ between fully online and in-person degree programs.

In this study, we examined student representation and grades in a fully online biology degree program compared to an equivalent in-person biology degree program offered at a large non-selective public university in the southwestern United States. Systematic disparities

in grades and other academic outcomes along the lines of gender, race/ethnicity and class reflect problems with institutional structures such as course and assessment design as well as broader social inequities, i.e. an "opportunity gap" [49]. In addition to looking at disparities in grades between students with different social identities, we also took a quantitative intersectional approach based on Black Feminist Theory developed by Kimberlé Crenshaw, Angela Davis, Combahee River Collective and Patricia Hill Collins among others [50–53] and asked how the interlocking systems of oppression of racism, classism and sexism could affect disparities in the grades that students receive. Several studies have shown that race/ethnicity, social class, and gender interact in a variety of ways to influence differences in grades and graduation rates among students with different social identities in college education [45, 54–57]. For example, Harackiewicz et al. found that the average grade disparity in an undergraduate biology course between first-generation BLNP (Black, Latinx, Native American, and Pacific Islander) students and continuing generation white and Asian students was about twice that between continuing generation BLNP students and white and Asian students [45]. Compounding effects of this nature or other forms of interactions might be observed in online courses as well.

### Research questions

**Research question 1.**   Has the online biology degree program increased representation of students with social identities that are underrepresented in STEM such as women, Black students, Latinx students, first-generation students, and low-income students compared to an in-person degree program?

**Research question 2.**   **(a)** Within the biology program's core required courses for all majors, do students with social identities that are historically underrepresented in STEM receive lower grades (women, Black, Latinx, first-generation, low-income) compared to students with social identities that are historically overrepresented in STEM (men, white, continuing generation, middle- to high-income)? **(b)** Do these grade gaps differ between the online science courses and the in-person courses? We define grade gaps here as the difference in grades received by students with social identities historically overrepresented in STEM compared to the grades received by students with social identities historically underrepresented in STEM. While women are no longer underrepresented at the undergraduate biology level, they still are underrepresented in the professoriate.

**Research question 3.**   Are grade gaps exacerbated for students holding multiple social identities that are underrepresented in STEM?

**Research question 4.**   Do grade gaps between students holding multiple identities that are underrepresented in STEM and students with social identities that are overrepresented in STEM differ between online and in-person courses?

## Methods

### Data

This study uses student course grades, student GPAs, and demographic data from students enrolled in in-person and fully online biology degree programs from a single non-selective university. All data were collected from the university registrar. The data were anonymized, and our analysis was performed in accordance with an institutional review board-approved protocol. We included data from students enrolled in the Biological Sciences major and from the set of core science courses required for that major. These included introductory biology courses, an evolution course, and a genetics course (six biology courses total); two general chemistry courses and two organic chemistry courses (four chemistry courses total); and two

general physics courses (two physics courses total). Course grades were taken from the Fall 2014–Spring 2018 semesters. This research was conducted under a protocol approved by the Arizona State University institutional review board (STUDY #9105). No consent was required because all data was analyzed anonymously.

The Biological Sciences major was chosen because it is a high enrollment program that is represented in both the in-person and fully online settings. The degree requirements for the in-person and online degrees are equivalent, and online and in-person courses have been backward designed so that they have similar learning goals/outcomes and course content. Many of the same instructors of in-person courses have designed the online version of the course. Although this unified design makes a comparison between courses in these two degree programs possible, it is important to acknowledge that while the courses are intended to be equivalent, there are likely to be some differences between them. A notable difference is that in-person courses are only offered over 15-week, whereas online courses are only offered over 7.5-weeks. Students are advised to take half as many courses in the 7.5 week session than the 15-week session, so even though the course pace of one course is accelerated, the overall course load should be the same. Given our study design, it was not possible to control for the assessment formats, grading schemes, or other factors that are left at the discretion of individual course instructors. Although we know that there are differences among instructors in their assessment decisions even in the same in-person course offering [58], we do not know of any systematic differences between online and in-person courses as far as assessments.

Course grades are on a 4.33 scale (4.33 = A+, 4.0 = A, 3.66 = A-,. . ., 0 = Fail). Students who received "withdraw" grades were converted to failing (0.0) grades. We made this choice because all of the courses in question are required for graduation, thus, withdrawing and needing to later retake the course has a similar negative effect on a student as failing the course even though withdraws do not impact student GPA. It is important to note that there are systematic differences in withdraw rates by demographic groups and between in-person and online courses (Table 1). More students withdraw from online sections (13.9% compared to 9.0%, $X^2(1) = 53.211$, $p < .001$), and, considering there is a shorter window of time to drop courses in the online program, any decision regarding treatment of withdraws will impact our overall conclusions. For interested readers, we provide a complete alternative analysis in the supplemental materials in which withdraws were excluded (S1–S6 Tables in S2 File).

We used grade point average in other courses (GPAO) to indicate each student's prior academic success [41, 59]. For any student-course enrollment in our dataset, we calculated GPAO

**Table 1. Withdraw percentages by student group and mode of instruction.**

| Demographic Category | Withdraw Percentage | |
|---|---|---|
| **Gender** | **Women** | **Men** |
| In-person | 8.2 | 8.9 |
| Online | 14.2 | 12.5 |
| Race/Ethnicity | BLNP | White |
| In-person | 10.1 | 8.5 |
| Online | 16.5 | 12.5 |
| College generation status | First-generation | Continuing generation |
| In-person | 10.4 | 7.6 |
| Online | 13.1 | 14.2 |
| Pell eligibility | Pell eligible | Non-Pell eligible |
| In-person | 10.2 | 7.2 |
| Online | 14.0 | 13.5 |

by removing the contribution of the course in question from the end-of-term GPA record. We preferred this method over using start-of-course GPA because it allows us to have an academic success indicator that is valid for first-semester students. Because this institution does not require incoming test scores, significant missing data prevented us from relying on high school GPA or standardized test scores as a proxy for academic achievement.

Demographic data were collected from the registrar. For race/ethnicity, we included only students identified as white, Black, Latinx, Native American, or Pacific Islander. We then operationalized this category to the majority group (white) and groups that are underrepresented in STEM, which include Black, Latinx, Native American, and Pacific Islander (which we abbreviate in the figures as BLNP). We prefer this abbreviation to the commonly used term URM (underrepresented minority), because it retains the names that people in these groups use to identify themselves [60, 61]. We purposely do not capitalize white because "white" does not represent a shared culture and history in the way Black does and has long been capitalized by hate groups [62]. Our analysis excludes Asian students, students reported as belonging to two or more races, and any students for whom these data are unreported or missing. We chose to exclude Asian students because our focus in this study is on groups that are historically underrepresented in science in the US and Asian students have not been included in the National Science Foundation's definition [26]. We acknowledge that there are many subsets of Asian students, some of whom are underrepresented in STEM in the US [63]. Unfortunately, we were not able to disaggregate Asian students because of the way that these data were collected by the institution. We also excluded the two or more races category due to uncertainty about group membership. For gender, we only included women and men in our analyses. This excluded two students who had non-binary genders recorded. We acknowledge that gender exists as a continuum, but we were not able to statistically analyze the two non-binary students. For income level, we included federal Pell grant eligibility. Students who received any Pell grant money in any academic year studied were classified as Pell eligible and considered low-income. All other students were considered non-Pell eligible. For college generation status, we used first-generation college student data collected from students' college application and financial aid documents. We categorized students as first-generation to college if neither parent had earned a college degree and as continuing generation otherwise. This resulted in a final data set of 10,249 student-course-enrollments across 24 courses (treating in-person and online versions of the same course as distinct courses).

## Data analysis

To assess if student representation in the online biology program was significantly different from the in-person program, we compared the proportions of students using a chi-square test. We used mixed effects linear modeling to determine the effects of four demographic variables (gender, race/ethnicity, college generation status, and federal Pell grant eligibility) as well as online modality in predicting course grades. We used GPAO to adjust for each student's general academic success. GPAO was centered separately among the in-person and online students, using each cluster's mean GPAO [64]. This centering choice follows from the fact that the two populations have no enrollment overlap—online course sections are open to only online students, and vice versa—and that there are systematic differences in how students are graded between the online and in-person modes. We also included random effects components on the model intercept. We used separate random effect terms to account for the data dependency (i.e., clustering) caused by (1) multiple student-enrollment records coming from the same students, and (2) individual records nested within course sections, respectively. We calculated intraclass correlation coefficients for our base model (model A below) with random

effects for both students and sections and compared it to models with only one of these random effects. The intraclass correlation coefficient (ICC, Eq 1) [65] is a ratio of the variance explained by random effects to the total variance explained by random and fixed effects plus residual variance. In model A (below) ICC was 0.356. Variants of model A including only one of these random effects show ICC values for course sections of 0.100 and for individual students of 0.236. These ICC values show that the random effects are appreciable and should be included in our modeling.

$$ICC = \sigma_\alpha^2 / (\sigma_\alpha^2 + \sigma_f^2 + \sigma_\varepsilon^2) \qquad \text{Eq 1}$$

We began with a regression that included the GPAO term and the four demographic terms. We examined each possible addition to the model in a stepwise manner and made model selections based on comparison of AIC values and statistical comparisons using likelihood ratio tests.

Statistical analyses were performed in R [66] and made use of the lme4 [67], lmerTest [68], MuMIn [69], performance [70], and sjPlot [71] packages for the modeling. The anonymized data used in this study as well as the R code used in our analyses are provided in S2 File.

## Results

### Finding 1: Compared to in-person program, online biology degree program increases representation of women, first-generation, and Pell eligible students, but not BLNP students

Women, white students, and continuing generation students are in the majority in both the in-person and online populations we studied. Non-Pell eligible students represent a slight majority of in-person students, whereas Pell eligible students are a slight majority online (Fig 1). A

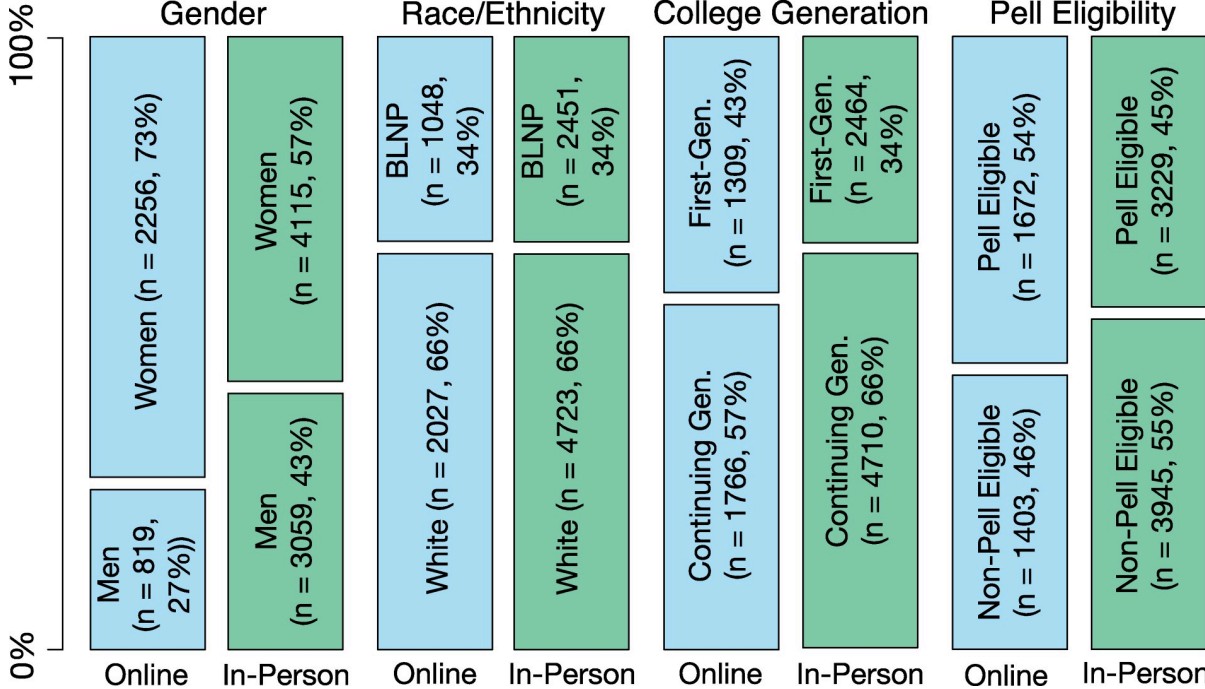

**Fig 1. Demographic comparisons of the online biology program compared to the in-person biology program.** BLNP refers to Black, Latinx, Native American, and Pacific Islanders. First-generation students are those for whom neither parent had earned a college degree. Pell eligible refers to low-income students.

summary of racial/ethnic data within the BLNP category and between in-person and online populations is provided in S7 Table in S2 File. Statistical analysis of the differences between in-person and online populations shows that the proportion of BLNP students between the in-person and online biology program is not statistically different ($X^2(1) = 0.00$, $p = .95$), while the proportions of women ($X^2(1) = 233.76$, $p < .001$), first-generation students ($X^2(1) = 62.21$, $p < .001$), and Pell eligible students ($X^2(1) = 75.27$, $p < .001$) are each significantly higher online compared to in-person (Fig 1).

## Finding 2: Women, first-generation, Pell eligible, and BLNP students receive lower grades regardless of modality. For first-generation students, this grade gap is eliminated in online courses

To assess whether students with historically underrepresented social identities in STEM receive lower grades in science courses than students with overrepresented social identities, we first ran a model with course grade as the outcome and demographic variables as predictors while controlling for student grades in other courses (see Table 2 for full model specification). This initial model (A), shows that women, first-generation, Pell eligible, and BLNP students are each associated with lower course grades (Tables 2 and 3). The size of these effects in model A ranges from around -0.08 grade units for women and first-generation students, to -0.11 for Pell eligible students, to -0.22 for BLNP students.

Next, in model B, we added the mode of instruction term (online or in-person). This significantly improved model fit (likelihood ratio test, $p < .001$) and showed that the online mode of instruction was associated with a large, negative grade difference. This means that on average, students earn lower grades online. Finally, to consider whether this online effect is differential by student demographics, we added interactions between mode of instruction and each demographic variable, in turn, in models C–F. Model D, which includes an interaction between first-generation status and mode of instruction, was the best-fitting model based on AIC and likelihood ratio tests (likelihood ratio test compared to model B, $p = .009$). According to this model, while first-generation students earn lower grades in-person than continuing generation students, there is almost no grade gap between first-generation and continuing generation students in online courses. Effects plots of model D are shown in Fig 2 and fit statistics are shown for all models in S8 Table in S2 File.

Although the inclusion of the first-generation status and mode of instruction interaction in model D was the only change among models C–F to significantly improve upon the no-interactions model (B), model F that includes an interaction between race/ethnicity and mode of instruction does at least marginally improve model B as well. Model F shows a positive interaction between BLNP and online instruction, analogous to the interaction effect in model D, i.e. the grade disparity between white and BLNP students is lower online than in-person. Although

**Table 2. Models compared for research question 2.**

| A: Course Grade ~ GPAO + Gender + College Generation + Pell Eligibility + Race/Ethnicity + (1 | STUDENT) + (1 | SECTION) | |
|---|---|
| B: | ~ {Model A} + Online |
| C: | ~ {Model B} + Gender: Online |
| D: | ~ {Model B} + College Generation: Online |
| E: | ~ {Model B} + Pell Eligibility: Online |
| F: | ~ {Model B} + Race/Ethnicity: Online |

**Table 3. Regression results for research questions 2a and 2b.** BLNP refers to Black, Latinx, Native American, and Pacific Islanders. First-generation students are those for whom neither parent had earned a college degree. Pell eligible refers to low-income students.

| | Model A | Model B | Model D |
|---|---|---|---|
| Intercept | 2.664*** (0.038) | 2.721*** (0.039) | 2.740*** (0.040) |
| GPAO | 0.643*** (0.015) | 0.644*** (0.015) | 0.643*** (0.015) |
| Gender (W) | -0.078* (0.032) | -0.066* (0.032) | -0.064* (0.032) |
| College Generation (FG) | -0.085* (0.033) | -0.081* (0.033) | -0.151*** (0.043) |
| Pell Eligibility (Y) | -0.109*** (0.032) | -0.104** (0.032) | -0.095** (0.032) |
| Race/Ethnicity (BLNP) | -0.215*** (0.034) | -0.215*** (0.032) | -0.212*** (0.033) |
| Online (Y) | | -0.500*** (0.078) | -0.571*** (0.082) |
| Online (Y): College Generation (FG) | | | 0.170** (0.065) |
| AIC | 30058 | 30019 | 30014 |
| R-squared$_{marginal}$ | 0.215 | 0.241 | 0.242 |
| No. Observations | 10249 | 10249 | 10249 |

Standard errors are reported in parentheses.

*, **, *** indicate significance at the 95%, 99%, and 99.9% levels, respectively.

it was not significant in our dataset, it is possible that this is a real effect and may merit further examination, especially with disaggregated race/ethnicity data. Interestingly, in alternative modeling with withdraw grades excluded, model F is the best fitting of models C–F (S2 File).

For readers interested in viewing the full dataset, S1–S4 Figs present simple linear regressions for each demographic subpopulation. Model fit statistics and results for all models are shown in S2 File. The full data and R code for analyses are also included in S2 File.

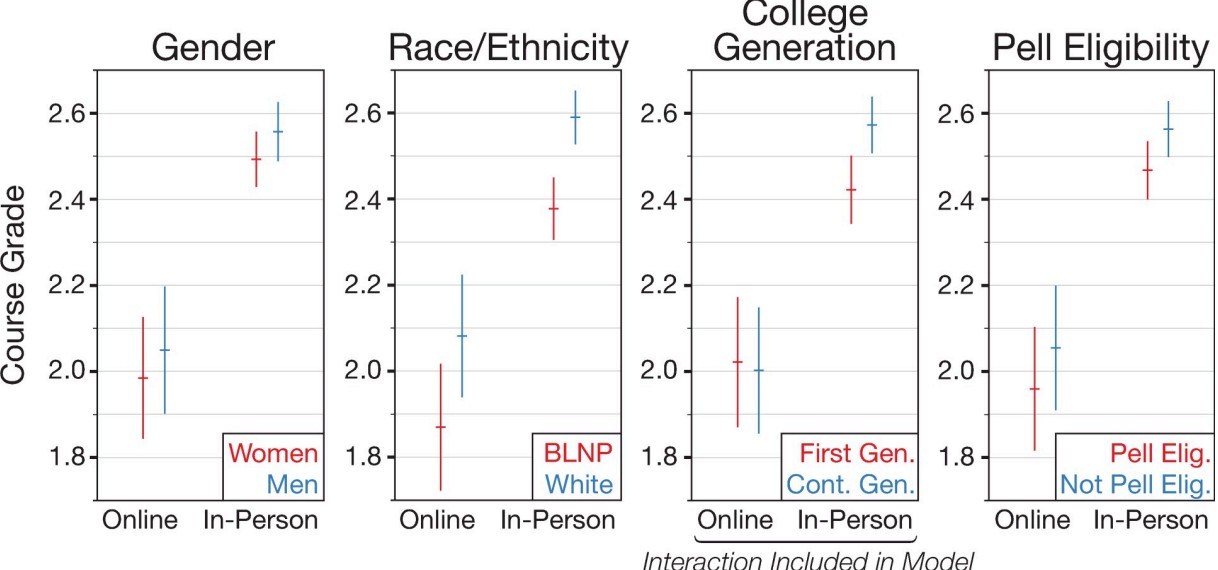

**Fig 2. Predicted demographic effects and interactions with online instruction mode based on model D.** Model D includes only the interaction involving college generation; the remaining interactions are shown for reference. Overall, model D finds a large and negative effect associated with online instruction and smaller negative effects associated with each of the underrepresented demographic groups. The significant interaction between generation status and online instruction shows that the effect of generation status is present only for in-person instruction. Error bars represent 95% confidence intervals. BLNP refers to Black, Latinx, Native American, and Pacific Islanders. First-generation students are those for whom neither parent had earned a college degree. Pell eligible refers to low-income students.

**Table 4. Models compared for research question 3.**

| | |
|---|---|
| A: Course Grade ~ GPAO + Gender + College Generation + Pell Eligibility + Race/Ethnicity + (1 | STUDENT) + (1 | SECTION) | |
| G: | ~ {Model A} + Pell Eligibility: Race/Ethnicity |
| H: | ~ {Model A} + College Generation: Race/Ethnicity |
| I: | ~ {Model A} + Gender: Race/Ethnicity |
| J: | ~ {Model A} + College Generation: Pell Eligibility |
| K: | ~ {Model A} + Gender: College Generation |
| L: | ~ {Model A} + Gender: Pell Eligibility |

## Finding 3: Grade disparities are generally not exacerbated for students holding multiple identities underrepresented in STEM

Beginning from the initial, no-interaction model from the previous section (model A), we tested additional pairwise interactions between the demographic terms (G–L, Table 4). None of these represent a significant improvement over model A. Of these, model G (Pell Eligibility: Race/Ethnicity) has the most support from AIC, offering some evidence that BLNP students who are Pell eligible receive lower grades than BLNP students who are not Pell eligible (S10 Table in S2 File). Other than BLNP and Pell eligibility, grade gaps were not exacerbated for students holding any combination of identities examined here.

## Finding 4: Continuing generation women students receive better grades than first-generation women in-person, but this difference is not observed online. Men receive higher grades than women online, but not as much in-person

Using model D (College Generation: Online) as a point of comparison, we again tested the inclusion of all possible pairwise interactions for the demographic categories, but here we also included an interaction between these pairwise combinations and instruction mode (Table 5). As with the findings from RQ3, we find that most of the models tested are similar or worse in terms of AIC (S8 Table in S2 File). The best fitting of these new models was Model Q, which includes a three-way "gender: first-generation: instruction mode" interaction (Table 6). This model indicates that the interaction between first-generation status and mode of instruction in model D is predominantly driven by the additional interaction with gender. There is little grade gap between first-generation and continuing generation men, both online and in-person (Fig 3C). However, continuing generation women receive higher grades than first generation women in in-person courses, but not in online courses (Fig 3C). This model also indicates that

**Table 5. Models compared for research question 4.**

| | |
|---|---|
| B: Course Grade ~ GPAO + Online + Gender + College Generation + Pell Eligibility + Race/Ethnicity + (1 | STUDENT) + (1 | SECTION) | |
| D: | ~ {Model B} + College Generation: Online |
| M: | ~ {Model B} + Pell Eligibility: Race/Ethnicity: Online |
| N: | ~ {Model B} + College Generation: Race/Ethnicity: Online |
| O: | ~ {Model B} + Gender: Race/Ethnicity: Online |
| P: | ~ {Model B} + College Generation: Pell Eligibility: Online |
| Q: | ~ {Model B} + Gender: College Generation: Online |
| R: | ~ {Model B} + Gender: Pell Eligibility: Online |

**Table 6. Regression results for models examining whether online modality exacerbates or reduces any grade disparities for students holding multiple social identities underrepresented in STEM.** BLNP refers to Black, Latinx, Native American, and Pacific Islanders. First-generation students are those for whom neither parent had earned a college degree. Pell eligible refers to low-income students.

| | Model D | Model Q |
|---|---|---|
| Intercept | 2.740\*\*\* (0.040) | 2.697\*\*\* (0.045) |
| GPAO | 0.643\*\*\* (0.015) | 0.641\*\*\* (0.015) |
| Gender (W) | -0.064\* (0.032) | 0.011 (0.049) |
| College Generation (FG) | -0.151\*\*\* (0.043) | -0.074 (0.065) |
| Pell Eligibility (Y) | -0.095\*\* (0.032) | -0.093\*\* (0.032) |
| Race/Ethnicity (BLNP) | -0.212\*\*\* (0.033) | -0.210\*\*\* (0.033) |
| Online (Y) | -0.571\*\*\* (0.082) | -0.454\*\*\* (0.101) |
| Online (Y): College Generation (FG) | 0.170\*\* (0.065) | 0.018 (0.118) |
| Online (Y): Gender (W) | | -0.178\* (0.089) |
| College Generation (FG): Gender (W) | | -0.133 (0.083) |
| Online (Y): College Generation (FG): Gender (W) | | 0.235† (0.141) |
| AIC | 30014 | 30015 |
| R-squared$_{marginal}$ | 0.242 | 0.243 |
| No. Observations | 10249 | 10249 |

Standard errors are reported in parentheses.

†, \*, \*\*, \*\*\* indicate significance at the 90%, 95%, 99%, and 99.9% levels, respectively.

there is an interaction between gender and mode of instruction. While all students receive better grades in in-person courses than online courses, this effect is particularly large among continuing generation women, meaning that continuing generation women receive lower grades than continuing generation men online (Fig 3D).

## Discussion

Among the students with social identities that are historically underrepresented in STEM, we found that all except BLNP students are enrolled in higher proportions in the online biology degree program compared to the equivalent in-person biology degree program. This speaks to one of the commonly discussed benefits of online degree programs: increased access. However, all of our regression models of course grade show that the online instruction mode is associated with lower course grades for all students after controlling for the other factors. This is also consistent with prior literature showing lower student grades in online courses, particularly at nonselective institutions. When employers and advanced degree programs are evaluating these students, particularly their GPAs, these lower course grades in core courses may limit their ability to maximally leverage their biology undergraduate degree. It is important to note that we only examined grade differences. Prior studies have indicated that there are other important disparities between fully online degree program students and students in an in-person degree program, including sources of career advice and access to high-impact practices such as undergraduate research [15]. Thus, increasing access to a degree is only the first step as these other inequities need to be considered.

The cause of the overall lower online grades is difficult to identify definitively within this study. Our finding is consistent with some prior work [11, 12, 33, 34], but not all [14, 32]. There are several factors that could be influencing the lower online grades in our study. Online instructors may be less sensitive to the specific concepts that students are struggling with, there may be fewer opportunities for formative feedback because of the modality, or students may

## Two-Way Interactions

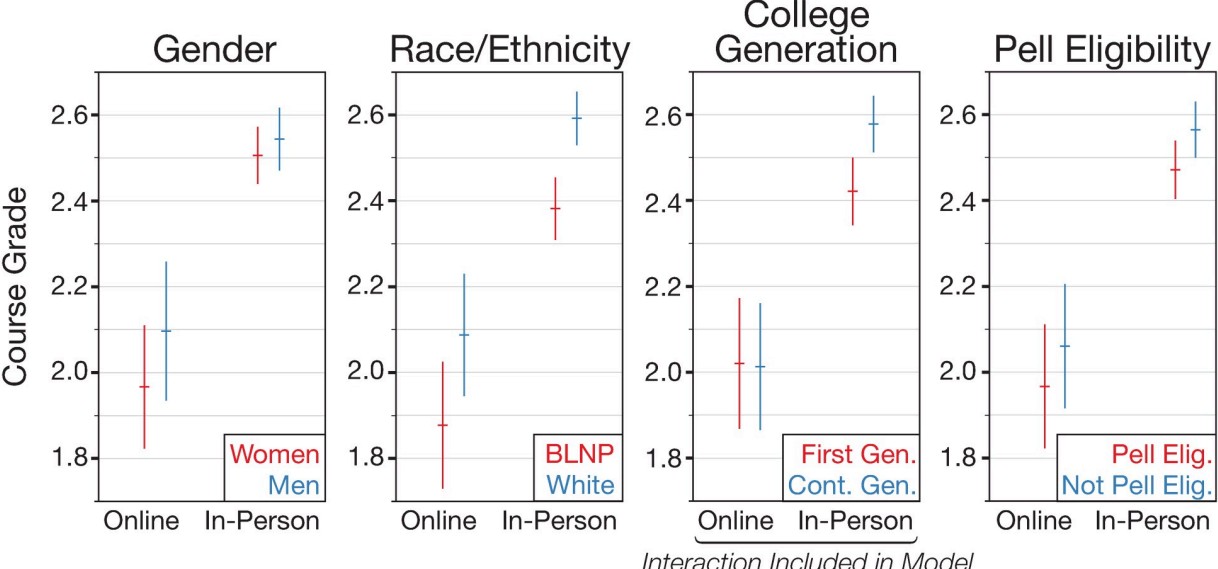

## Three-Way Interaction

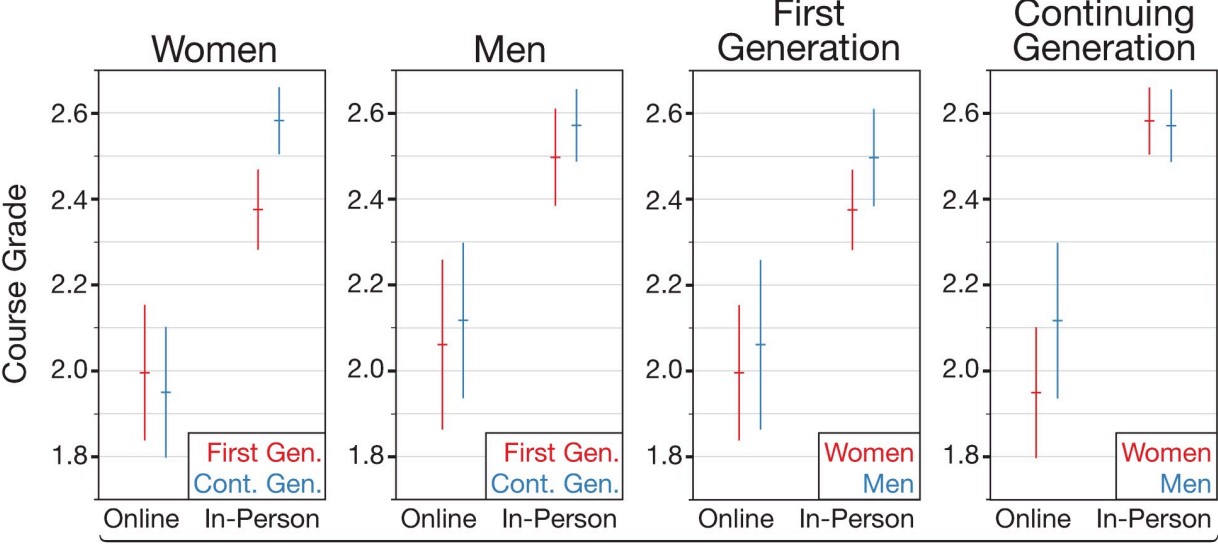

**Fig 3. Predicted demographic effects and interactions with online instruction mode based on model Q.** Model Q includes the two- and three-way interactions involving college generation; the remaining interactions are shown for reference. As with model D (Fig 2), model Q finds a large and negative effect associated with online instruction and a significant positive interaction for first-generation students online. The added three-way interaction shows that the interaction between college generation and online instruction is seen most strongly among women. Error bars represent 95% confidence intervals. BLNP refers to Black, Latinx, Native American, and Pacific Islanders. First-generation students are those for whom neither parent had earned a college degree. Pell eligible refers to low-income students.

struggle with building relationships with the instructor. In addition, in our study specifically, the shortened 7.5 week terms for the online courses mean that a student who suffers an illness, experiences a major life event, or is otherwise prevented from doing coursework for several days or a week will fall disproportionately further behind than a student who is enrolled in a

15 week term. Finally, although our models do account for the students' college performance, they do not fully control for academic preparation, thus the online effect could be partially explained by lesser college preparedness. This would be consistent with Cavanaugh & Jacquemin (2015) who, while finding a negligible difference in online vs. face-to-face courses, also found that online courses tend to magnify the effect of differences in GPA, with low GPA students performing even worse and high GPA students performing even better [14, 32].

Regarding the specific demographic effects on course grades and considering only main effects, we found that women, first-generation students, low-income students eligible for federal Pell grants, and BLNP students are more likely to receive lower course grades than men, continuing generation students, middle- to high-income students, and white students both online and in-person. So, the online format does not seem to eliminate grade gaps for these populations historically underrepresented in STEM undergraduate degree programs. However, by considering differential effects of online instruction on demographic groups, we found that the grade gap between first-generation and continuing generation students differs between the online and in-person degree programs. While continuing generation students receive better grades than first-generation students in in-person courses, grades are lower for both groups online such that there is almost no grade difference between these two groups in online courses.

By also considering interactions of social identities, we were able to point to a more specific explanation of this interaction between first-generation status and online instruction. When we added an additional interaction with gender (model Q), we observe the significant first-generation and online interaction only among women. Thus, first-generation women earn similar grades compared to continuing generation women in the online degree program, whereas first-generation women *in-person* earn lower grades than their continuing generation peers. This model also shows an interaction effect between gender and online instruction, suggesting that, overall, women receive lower grades online compared to men.

## Explanations

Our results show considerable similarity between in-person and online degree program courses in that BLNP students and students eligible for federal Pell grants received lower grades compared to white students and students that are not eligible for Pell grants in both in-person and online degree programs. This is consistent with prior literature comparing online and in-person STEM courses such as Wladis et al. [30] whose modeling found neither race/ethnicity nor Pell grant eligibility to significantly explain online course grades. Therefore, it is quite possible that the same institutional and societal mechanisms that are hypothesized to drive grade gaps in-person have a similar effect online. One of the most common suggestions for lower exam scores of BLNP students in college science courses is stereotype threat—the phenomenon wherein individuals from negatively stereotyped groups feel pressured to perform well lest they be seen as fulfilling those stereotypes [30, 72, 73]. Although our results do not indicate, as Wladis and coauthors suggest, that stereotype threat is more detrimental online, we still consider it to be a possible factor in explaining the observed grade differences. One reason for similar levels of stereotype threat in online and in-person science courses could be similar prevalence of high-stakes assessments such as timed exams which are known to exacerbate stereotype threat [73, 74]. Reducing the contribution of exams toward final grades and providing frequent low-stakes assessments have been shown to reduce grade gaps in STEM courses [46, 75]. Lastly, given the larger societal context in the United States, structural racism and large opportunity gaps by race and social class are additional possible reasons for persistent grade gaps in online and in-person science courses by race/ethnicity and Pell

grant eligibility. In other words, our finding of similar demographic grade differences surely derives in part from the fact that the online and in-person degree programs both operate within a society that is highly unequal (particularly with regard to academic preparation and opportunities) and in part due to institutional structures and traditions such as high-stakes timed exams that exacerbate these inequalities.

Although our results suggest in-person and online equivalence in grade gaps with respect to race/ethnicity and Pell grant eligibility, our models show that women receive lower grades than men in online courses but this grade gap is smaller in-person courses. Previous studies focused on the performance of women in in-person science courses may help explain why women experience larger grade gaps online. Eddy & Brownell [42] reviewed the literature of gender disparities in undergraduate STEM education and reported widespread gender gaps in academic performance, engagement, and affective measures such as self-efficacy, sense of belonging, and science identity. They describe four sociocultural factors underlying the observed gender gaps: stereotype threat, implicit or explicit biases held by peers and/or instructors, conflict between personal goals and stereotypes about STEM professionals, and implicit theories of intelligence. As we suggested above, stereotype threat is a plausible driver of gaps in both instructional modes. However, classroom interactions may also be a factor. Eddy et al. [47] observed substantially lower classroom participation in-person among women compared to men. Subsequent studies in the context of in-person courses have documented gender differences in student comfort participating in front of the whole class [76], lower likelihood of women participating in larger classes [77], student perception of their own intelligence when comparing it to others [78], and preference for working with a friend in small group work [76]. Such differences not only represent missed learning opportunities for women to participate but can also perpetuate stereotypes about men being smarter or more capable in science. The effect of class participation is difficult to predict. Given that class participation in online courses often involves posting on a discussion board that is visible to the full class, many of the same biases and social pressures discussed by Eddy et al. and Ballen et al. as far as whole class discussions may be present in online discussion boards too. On the other hand, some studies show that women participate disproportionately more in web-based classroom discussions [79, 80]. Finally, student age data show that in our dataset, men enrolled in online degrees are on average about two years older than women. This may imply other differences in life circumstances between men and women in online programs. Thus, whereas for traditional-aged students, women typically come to college with better preparation [81], it is possible that within the population interested in pursuing online degrees, men are more prepared. Lastly, in explaining their finding that women performed worse in online courses, Wladis et al. [30] suggest that women taking online courses are more likely to be primary caregivers of small children and thus have less time available for coursework, which is consistent with a previous study of a fully online biology degree program [15].

The second difference we find with respect to mode of instruction is a reduction of the grade gap for first-generation students in the online setting, particularly among women. Prior work studying the success of first-generation to college students in the context of in-person courses suggests a range of factors that may drive lower grades compared to continuing generation students, including general academic preparation, poverty, and feeling unconnected to or threatened by the college environment [82]. Model D which includes an interaction between online modality and first-generation status shows that the first-generation grade gap is eliminated in the online instruction mode, while model Q which includes a three-way interaction between gender, online modality and first-generation status shows this reduction being limited to first-generation women online. This could reflect that the online college experience is in fact equivalent for first-generation and continuing generation students. Because online

college is a relatively novel experience, one possibility is that there is no advantage from having a family history in college. Alternatively, because first-generation students make up a larger percentage of online students, these students may perceive the online program to be more welcoming and may not feel out of place in the way that some first-generation students feel in in-person settings when they interact with other students. A different possibility is that students are not interacting with each other as much online, so there are fewer opportunities for them to compare themselves to each other and not feel as though they belong. Finally, instructors may be more explicit with the norms and expectations of online courses than they are with in-person courses, so information about the course may be more equitably distributed to students through a syllabus or online course materials. However, it is important to note that the lack of grade gap in the online modality is due to poorer performance of continuing generation students online, not because of better performance of first-generation students online.

Having discussed possible explanations for these effects, it is important to explain that these estimates represent a conservative lower bound on the in-practice demographic effects on college course grades. There are two reasons why this is the case, which we will discuss in turn along with alternative models that provide possible estimates of the true effects (S11 Table in S2 File).

### Alternative subgroup-centered GPAO model

The first, and most significant, reason that our estimates are conservative is that there are systematic differences in GPAO between the demographic groups studied and the demographic compositions are systematically different across instruction mode. Because of these systematic differences and because our models used GPAO centered within each of the in-person and online instruction modes, the estimated demographic effects are smaller than those that would be found by simply calculating average grades by group. To estimate the maximum variance that can be ascribed to demographics, we can run alternative models in which the GPAO term is centered within both instruction modes and demographic subgroups. In effect, this forces all of the between-group variation onto the demographic terms. In this alternative model, the explanation for the demographic effects includes both the pre-college academic preparation differences and any university policy, culture, or classroom causes.

We ran two subgroup-centered models (S and T, S12 Table in S2 File), based on models D (College Generation: Online) and Q (Gender: College Generation: Online), respectively. As expected, these yield larger estimates of the demographic effects and interaction effects than our main models. It is also notable that for these subgroup-centered models, the more complex model with a three-way interaction between gender, first-generation status, and online instruction mode (T) is a significantly better fit than model S, which only has a two-way interaction between first-generation status and online instruction mode (likelihood ratio test, $p < .001$). This adds credibility to the effects seen in the main model Q, which was a non-significant improvement over model D. That is, it provides further evidence that the grade gap between first-generation and continuing generation is lower among women in the online program compared to in-person program, but there is little grade gap between first-generation and continuing generation men in both the online and in-person programs.

### Alternative prior GPA model

The second reason our estimates are conservative stems from our use of students' college GPAs to account for overall academic performance. We chose to use college GPA because measures of pre-college academic performance, such as high school GPA or college transfer GPA, suffer from significant missing data for our student population in ways that may skew

the data. However, when we run our models on the subset of data for which high school GPA or college transfer GPA are available, we find slightly larger effects that are otherwise consistent with our main model results (S13 Table in S2 File). Detailed results from these models are reported in S2 File.

The most likely explanation for this difference in magnitude is that students from these demographic groups receive lower grades in all of their courses, thus by using GPAO we are somewhat muting the true demographic effects. That said, the general concordance of results from this alternative model lends support to our overall conclusion that there are systematic demographic grade gaps in these science courses. We chose to be conservative in our approach presented in the study, but the real impact on students is likely larger than our estimates.

## Limitations

All of our main models use GPAO to adjust for a student's general academic success. However, because in-person degree students cannot enroll in the same courses as online degree students, and vice versa, we cannot test the consistency or similarity of grading between these two modes. This somewhat increases the uncertainty of our model estimates for the effect of instruction mode. However, the equivalent in-person and online courses are offered by the same academic units, so we assume that the grading policies and student expectations are similar in all offerings.

Previous studies have included student age as a predictor of online course grades. We did examine student ages within our dataset and found the expected pattern of online students being substantially older than in-person students ($M_{online}$ = 27.3 yrs; $M_{in-person}$ = 23.4 yrs; t. test = 32.425, $p < .001$). There were also smaller age differences (±1 year) between demographic groups within each instruction mode population. The largest of these showed that men in the online degree program were 2.1 years older, on average, than women. Because the age difference is so strongly correlated with instruction mode, it was not feasible with the available data to separately estimate predictive effects due to student age and instruction mode.

As described in the methods section, we treated withdraw grades as fails, but we also noted that withdraws are more common in the online courses (Table 1), a pattern also seen in prior research [29, 30]. One possible cause for this difference within our particular dataset stems from differences in the academic schedule. All of the studied online courses are taught in abbreviated 7.5-week terms. As a result, students have a very short window during which to drop a course and avoid having it appear on their transcript. Beyond that time, students must either complete the course for a conventional letter grade or withdraw. Other possible causes include differences in academic advising or differences in academic preparation. To explore the degree to which the treatment of withdraws alters our findings, we ran alternative models corresponding to models A–Q on a dataset with withdraws excluded instead of counted as failures (see S2 File). These models show reduced, but still significant and large, negative online effects. The models also differ somewhat in the magnitude of the demographic effects. These differences can be explained by the overall greater withdraw percentage in the online population and how withdraw percentages vary among the demographic groups studied (Table 1). Thus, the differences in withdrawal rates imparts some uncertainty to our results, but we do not feel it detracts from our overall conclusions. This analysis does highlight the complexity inherent in studying academic outcomes and the importance of exploring multiple outcome measures, where appropriate.

Another limitation of our study is that we did not disaggregate BLNP students due to low sample sizes, even though people with these racial/ethnic identities have vastly different histories and experiences in the US. However, these groups share the experience of being

underrepresented in STEM in the US, which is the specific context we examined in our study and was our rationale for aggregating them and comparing them to white students. Finally, we acknowledge that Pell grant eligibility is a crude measure of socioeconomic status and does not allow us to gain a detailed understanding of the effects of social class on student experiences in STEM courses. However, it was the only indicator of socioeconomic status we could access from the university registrar. We encourage future studies to survey and interview low-income students to further probe their experiences and factors that may influence their experience in online degree programs.

### Future directions

There are a number of points within this discussion that would benefit from additional, targeted research. Because this study used only administrative data, it was not possible to speak to or control for the many ways that a student's life circumstances, resources they could access, academic opportunities, or personal motivations could impact course grades. Similarly, institutional structures and traditions that shape individual instructors' course and assessment design could contribute to the patterns we have observed here. Our previous work in one in-person biology course showed that cognitive difficulty and format of exams were associated with gender and socioeconomic grade disparities [43]. Follow-up research should be undertaken that examines the effects of course design and assessment practices on grade gaps systematically or even experimentally while maintaining the focus on fully online degree programs. Whereas a number of studies exist that examine online learning broadly, fully online degree programs represent a new and important area of growth—one with potential to reduce historical inequities in STEM and higher education more broadly, but only if the institutional deficits presented here are addressed.

### Conclusions

Our results show that a fully online biology degree program increases access to college biology degree programs for women, first-generation students, and students eligible to receive federal Pell grants. However, it does not increase access for BLNP students, i.e. students that identify as Black/African American, Latinx/Hispanic, Native American/Native Alaskan or Pacific Islander/Native Hawaiian. Our results also show that gender, racial and socioeconomic disparities in grades received by students in science courses persist in both in-person and online courses. More specifically in our main analyses, BLNP students, first generation students, and students eligible for Pell grants, receive lower grades than white, continuing generation, and students not eligible for Pell grants, in both online and in-person modalities. Women receive lower grades than men in online courses, but not as much in in-person courses. However, for first-generation college students, particularly first-generation women, online instruction is associated with a complete reduction in the grade gap seen in the context of in-person instruction.

We hope that these results will encourage a continued effort to address the outcome inequalities that clearly exist for students with different social identities in both in-person and online programs. All colleges and universities should strive to provide inclusive excellence and access is only the first step. While online instruction can mitigate some of these access issues, there are still inequities online. Further, our alternative modelling shows how adjusting for prior academic performance may minimize the true size of outcome inequalities. In particular, we observe a much larger grade gap for women in online programs in this alternative model. Although this gap is probably driven by pre-college differences, it is important to consider what colleges and universities could do to help reduce the impact of pre-college differences themselves.

As more degree programs are moved online and as these programs continue to attract women, first-generation students, and students from low-income households, any disadvantage felt by online students will disproportionately disadvantage these students. Worse yet, this is potentially a hidden problem to these students if the programs are advertised as being truly equivalent or if they are being presented as ways to close opportunity gaps. To fully reach their potential in making college degrees attainable by all qualified students, online programs should carefully monitor such disparities in academic outcomes and be proactive in working to eliminate them.

## Supporting information

**S1 File. Data and analysis codes.**
(ZIP)

**S2 File.**
(DOCX)

**S1 Fig. Course grades vs. GPAO by gender and online status.**
(EPS)

**S2 Fig. Course grades vs. GPAO by race/ethnicity and online status.**
(EPS)

**S3 Fig. Course grades vs. GPAO by college generation and online status.**
(EPS)

**S4 Fig. Course grades vs. GPAO by Pell eligibility and online status.**
(EPS)

## Author Contributions

**Conceptualization:** Chris Mead, K. Supriya, Ariel D. Anbar, James P. Collins, Paul LePore, Sara E. Brownell.

**Data curation:** Chris Mead.

**Formal analysis:** Chris Mead, K. Supriya, Yi Zheng.

**Funding acquisition:** Ariel D. Anbar, James P. Collins, Paul LePore, Sara E. Brownell.

**Investigation:** Chris Mead, K. Supriya.

**Methodology:** Chris Mead, K. Supriya, Yi Zheng.

**Project administration:** Chris Mead, K. Supriya.

**Supervision:** Sara E. Brownell.

**Validation:** Chris Mead, K. Supriya, Sara E. Brownell.

**Visualization:** Chris Mead, K. Supriya.

**Writing – original draft:** Chris Mead, K. Supriya.

**Writing – review & editing:** Chris Mead, K. Supriya, Yi Zheng, Ariel D. Anbar, James P. Collins, Paul LePore, Sara E. Brownell.

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
