## [Decision Letter · Decision Letter 0]

3 Sep 2020

PONE-D-20-22633

Online biology degree program broadens access for women, first-generation to college, and low-income students, but grade disparities remain

PLOS ONE

Dear Dr. Mead,

Thank you for submitting your manuscript to PLOS ONE. After careful consideration, we feel that it has merit but does not fully meet PLOS ONE’s publication criteria as it currently stands. Therefore, we invite you to submit a revised version of the manuscript that addresses the points raised during the review process.

ACADEMIC EDITOR: Please insert comments here and delete this placeholder text when finished. Be sure to:

Indicate which changes you require for acceptance versus which changes you recommendAddress any conflicts between the reviews so that it's clear which advice the authors should followProvide specific feedback from your evaluation of the manuscript

We look forward to receiving your revised manuscript.

Kind regards,

Amy Prunuske

Academic Editor

PLOS ONE

Journal Requirements:

Additional Editor Comments (if provided):

The authors complete an important analysis comparing outcomes between online and face-to-face instruction comparing different student identities. The reviewers are favorable about the authors' analysis and offer several suggestions to improve the manuscript. One area that I would urge the authors to address is a better description of the online vs in person course elements that might be contributing to these outcomes. It is critical that biology educators understand what practices are leaving some students behind. For example consider the work of Joe Feldman Grading for Equity. I look forward to seeing a revised version of this work.

Reviewers' comments:

Reviewer's Responses to Questions

**Comments to the Author**

1. Is the manuscript technically sound, and do the data support the conclusions?

Reviewer #1: Yes

Reviewer #2: Partly

2. Has the statistical analysis been performed appropriately and rigorously? 

Reviewer #1: Yes

Reviewer #2: No

3. Have the authors made all data underlying the findings in their manuscript fully available?

Reviewer #1: Yes

Reviewer #2: Yes

4. Is the manuscript presented in an intelligible fashion and written in standard English?

Reviewer #1: Yes

Reviewer #2: Yes

5. Review Comments to the Author

Reviewer #1: The authors use rigorous statistical analysis to compare the ability of a fully-online biology degree program to improve access and grade disparities for groups traditionally underrepresented the life sciences. The authors are uniquely positioned to undertake such a study, having access to Registrar data at Arizona State University, one of the only universities in which parallel fully-online and in-person biology degree programs are available. Overall, the authors have done a good job at presenting a thorough analysis to support their conclusions of the ability of the fully-online curriculum to address traditional concerns to equity and inclusion in life science education. The following are questions and suggestions that will hopefully help the clarity and overall strength of the manuscript:

-Throughout the manuscript, the authors cite the fact that students in online courses tend to get lower grades than their in-person counterparts and that such disparities do not, in large part, seem to be corrected by the fully-online curriculum. Left unsaid are the assessment techniques that are used in online versus in-person courses and the instructors who are doing the evaluating the students in each modality. Clearly, these are considerations that have been looked at in other studies and are not the direct subject of this work. However, it would be helpful when considering the reasons behind grade disparities to have a better sense of the role of particular assessment tools in generating these disparities. In short, how is grading done in each modality, who is doing the grading, and how to these contribute to the disparities on which this paper is focused?

-The authors reference the fact that online course are offered in 7.5-week sessions as opposed to the 15-week offerings of the in-person format. This would suggest an accelerated pacing to the online courses that could play a significant role in the grade disparities that are observed between what is seen in the two modalities. It would be good to hear more about the author's perspective on the potential role of course pacing on student performance.

-The authors choose to treat students who withdraw from the courses as having failed the courses and provide extensive rationale for their choice while acknowledging potential limitations to their analysis. My concern stems from the fact that withdrawal from a course can be the result of several unpredictable pressures on a student's life that have nothing to do with poor course performance. Would simply removing withdrawing students from the data set eliminate some of the analytical problems that this cohort introduces to the study?

-In Table 1, expressing the results as percentages in an additional column would make the results easier to digest.

-In Tables 2, 4, and 5, it is difficult to follow the progression of the models presented by what is in the Table itself. For each, a Legend that highlights the differences between each model would be appreciated. This would be redundant to the text of the paper but make the interpretation of each Table as one reads the manuscript markedly easier.

-On a related note, in several spots in the text (lines 339-45, 510-13, 545, 551-2), the authors present their Models solely by their single letter identification. It is difficult to recall during the reading which models are which. For clarity, adding the feature of the model in parentheses after the model name would be appreciated.

-Lines 357-65 and 410-418 appear to be the Figure Legends for Figures 2 and 3 embedded into the text. These, clearly, will need to be moved to the actual location in which these Figures will be placed (Disregard if this placement was done at the direction of the journal).

-Lines 51-53 introduce the concept of the "education desert". Some additional language for why these deserts exist would help contextualize the argument set-up here.

-In lines 158-64, the authors provide citations describing improved confidence and personal value in science amongst students taking online classes. Some additional language describing why this is so would help contextualize the argument being set-up here.

-Line 231: place "white" in parentheses.

I hope that these comments will help to produce a final paper that more effectively conveys its important messages.

Reviewer #2: Overview

This paper describes the analysis and interpretation of student achievement as well as demographic and socioeconomic data from an institution of higher education’s registrar’s office. It provides insights into student achievement in an online and face-to-face biology degree program based on students’ “social groups”. The manuscript is well-organized, with clear research questions and a results and discussion section formatted around those questions, and well written. It contains important and novel results that would be of interest to biology educators and program administrators and thus merits publication with some revisions.

Suggestions for improvement or changes

Introduction

The introduction thoroughly demonstrates the potential for online courses to provide access and promote diversity in science. It also reviews literature indicating that students often perform worse in online courses and demonstrates that the data analyzed in the manuscript are unique in that they are results from students enrolled in an online or face-to-face STEM program.

However, the introduction does not describe what is known about how students in the social groups examined typically perform in a face-to-face classroom. Since the title mentions grade gaps remaining, it would be good to establish what is known about the grade gaps in face-to-face courses for these groups.

The term grade gaps in general is central to 3 out of 4 research questions asked and needs to be defined and examined more thoroughly in the introduction. I also want to warn the authors that this reference to “gaps” can be confusing. For example, on line 481 the authors cite an article that describes some reasoning for gender gaps in academic performance. When stated like this it is not clear which gender is expected to perform better and which is expected to perform worse. I felt the same confusion various times with the term in reference to the authors’ own results (i.e. lines 367,379-381). The language in the abstract with reference to higher or lower grades is more precise.

Methods

The first sentence of the methods is confusing. The authors used data from students enrolled in both fully online and face-to-face biology programs.

I disagree with the authors’ decision to score withdraw grades as Fs when calculating the students’ GPAs and this is my basis for answering "no" to question 2. That is not how GPAs are calculated. I did see the note in the discussion about the completion of an analysis that did not score Ws as Fs in the GPA calculation and believe that these are the data that should be presented in the paper. It is well known that withdraw rates are higher in online courses. To count them as Fs is likely having a larger impact on your measure of student achievement (GPA) for online students. Additionally, an earned F from a student who continued to invest time into a course is a different very outcome than choosing to withdraw from a course. Finally, the differences in the withdraw rates by course modality and social groups could be meaningful and may warrant further analysis and interpretation.

I understand the rationale provided in the paper that students who receive a W in a course are not making progress toward their degree and that is an important consideration. Perhaps it deserves another type of analysis that looks at which students are successful (this definition varies, but I typically see that defined as earning grades A-C) compared with those who are not (a category which would include the students who withdrew and received failing grades).

Results

In general, the results section is focused on the results of the statistical tests. It would improve the clarity of the paper if the authors also included descriptions of the data. For example, the written description of the results in Table 1 is very brief. It could be improved by describing the proportions or differences between the proportions for the groups of students. The same is true for the data demonstrating grade gaps. There is a lot written about the models, but not about the differences in the calculated GPAs for the groups of students.

Discussion

There is an incongruence between the results and the interpretation of the results in the discussion. This was my reasoning for answering "partly" to question 1. For example, the authors found similar grade gaps for many students who are underrepresented in the STEM fields regardless of the modality and clearly state that the online format did not reduce grade gaps (line 440). However, some of the studies used to interpret these finding are explanations for why greater gaps may exist in online courses (i.e. line 467-471). The interpretation makes it sound like your data confirmed larger grade gaps in the online format. In general, I recommend a more thorough interpretation of the findings that BOTH modalities are leading to grade gaps for groups underrepresented in STEM.

Additionally in the second paragraph under the “Explanations” heading, I also sense an incongruence between the results and interpretation. This paragraph is interpreting the finding that grade gaps were larger for women in the online classroom. Yet the evidence provided in lines 479-494 is not specific to the online environment.

The discussion could benefit from emphasizing the need for online and face-to-face biology educators to reflect on how their practices are leaving some of their students behind. This message may be even more important for online educators since they likely have more students from social groups that are underrepresented in STEM fields. There is some of this language in the last two paragraphs of the discussion, but the paper may have broader appeal if it provided some resources on best practices for creating an inclusive environment that helps all students learn in online and face-to-face classrooms.

6. PLOS authors have the option to publish the peer review history of their article (what does this mean?). If published, this will include your full peer review and any attached files.

Reviewer #1: **Yes: **Michael J. Wolyniak

Reviewer #2: No

---

## [Author Response · Author response to Decision Letter 0]

19 Oct 2020

Please refer to the Response to Reviewers document in the attached files.

---

## [Decision Letter · Decision Letter 1]

13 Nov 2020

PONE-D-20-22633R1

Online biology degree program broadens access for women, first-generation to college, and low-income students, but grade disparities remain

PLOS ONE

Dear Dr. Mead,

Thank you for submitting your manuscript to PLOS ONE. After careful consideration, we feel that it has merit but does not fully meet PLOS ONE’s publication criteria as it currently stands. Therefore, we invite you to submit a revised version of the manuscript that addresses the points raised during the review process.

ACADEMIC EDITOR:

The resubmitted manuscript addresses all of the reviewers' concern. 

Reviewer 3 has a few minor suggestions that I would like you to review prior to acceptance. 

I look forward to seeing the updated manuscript.

We look forward to receiving your revised manuscript.

Kind regards,

Amy Prunuske

Academic Editor

PLOS ONE

Reviewers' comments:

Reviewer's Responses to Questions

**Comments to the Author**

1. If the authors have adequately addressed your comments raised in a previous round of review and you feel that this manuscript is now acceptable for publication, you may indicate that here to bypass the “Comments to the Author” section, enter your conflict of interest statement in the “Confidential to Editor” section, and submit your "Accept" recommendation.

Reviewer #1: All comments have been addressed

Reviewer #3: All comments have been addressed

2. Is the manuscript technically sound, and do the data support the conclusions?

Reviewer #1: Yes

Reviewer #3: Yes

3. Has the statistical analysis been performed appropriately and rigorously? 

Reviewer #1: Yes

Reviewer #3: Yes

4. Have the authors made all data underlying the findings in their manuscript fully available?

Reviewer #1: Yes

Reviewer #3: Yes

5. Is the manuscript presented in an intelligible fashion and written in standard English?

Reviewer #1: Yes

Reviewer #3: Yes

6. Review Comments to the Author

Reviewer #1: (No Response)

Reviewer #3: In summary, the manuscript addresses a key research area comparing the access and equity as it relates to online vs in-person instruction. The previous review has resulted in significantly addressing a few areas of concern and hence, improving the overall readability of this work. I am also in agreement with the authors about the concern raised by the previous reviews as it relates to considering the DFW rates as failures in the regression model. The authors address it really well.

I am recommending some minor revisions based upon my reading and they are summarized as follows:

-Introduction: If any references related to existing grade disparities could be reported in the context of “intersectionality”? Cite references related to “Black Feminist Theory” as it relates to biology education studies. Not sure if the authors are pursuing this for the very first time in the given context and hence, "novel".

-Add a marker at the bottom of the X-axis to indicate the axis does not start at 0 (or whatever the lowest GPA recorded is-- even if it not 0 , it is most likely not 1.6 and therefore should have a small mark to indicate this) – please see figures 2 and 3.

-Add a citation/weblink from NSF to support line 294

-Recommend tabling models and regressions results – see lines 434, 466, 469

-Recommend adding some elaboration on how results are consistent with some but not all prior reported work – line 496

Besides, these recommendations, I do not find any inconsistencies in the overall manuscript. It is very well written and would recommend it for publication.

7. PLOS authors have the option to publish the peer review history of their article (what does this mean?). If published, this will include your full peer review and any attached files.

Reviewer #1: **Yes: **Michael J. Wolyniak

Reviewer #3: No

---

## [Author Response · Author response to Decision Letter 1]

25 Nov 2020

Please see uploaded response to reviewers document.

---

## [Editor Report · Decision Letter 2]

1 Dec 2020

Online biology degree program broadens access for women, first-generation to college, and low-income students, but grade disparities remain

PONE-D-20-22633R2

Dear Dr. Mead,

We’re pleased to inform you that your manuscript has been judged scientifically suitable for publication and will be formally accepted for publication once it meets all outstanding technical requirements.

Kind regards,

Amy Prunuske

Academic Editor

PLOS ONE
---

## [Editor Report · Acceptance letter]

3 Dec 2020

PONE-D-20-22633R2 

Online biology degree program broadens access for women, first-generation to college, and low-income students, but grade disparities remain 

Dear Dr. Mead:

I'm pleased to inform you that your manuscript has been deemed suitable for publication in PLOS ONE. Congratulations! Your manuscript is now with our production department. 

Kind regards, 

on behalf of

Dr. Amy Prunuske 

Academic Editor

PLOS ONE